# How can health systems sustain lessons drawn from emergency contexts? Evidence from Colombia

**Simon Turner** [ID]**\*, Mary Ruth Guevara Maldonado**

School of Management, University of los Andes, Bogotá, Colombia

\* s.turner@uniandes.edu.co

## Abstract

The Covid-19 pandemic demanded rapid adaptation to health systems internationally, but little is known about the sustained value of the approaches to learning adopted. How does ongoing environmental turbulence influence the lessons drawn from health system responses to pandemics? To address this question, we engage with, and further develop, Levitt and March's highly cited perspective on experiential learning by analyzing Colombian healthcare professionals' experiences gathered during semi-structured interviews. Interviews included representatives of national government, service providers, administrative staff, clinicians, including physicians and nurses, professional associations, and academics. Aspects from the macro, meso, and micro contexts associated with the sustainability of organizational learning were identified. At the macro level, reform efforts seem to overlook lessons learned from the pandemic and divert the attention of key actors. At the meso level, leadership uses success stories to motivate teams, but financial challenges and absence of formal evaluations hinder the sustainability of innovations. At the micro level, there is a diminished workforce capacity, some concerns about virtual professional training and difficulties to address mental health issues is difficult due to stoic professional identities, unrelenting tempo of medical work, and institutional encouragement. This study extends Levitt and March's organizational learning framework to include environmental turbulence as a factor influencing learning. It highlights that a turbulent context simultaneously triggers learning processes while being a precursor to the interpretation of experiences. The research concludes there are additional moderating variables for organizational learning like human resource capacities, political cycles, and infrastructure continuity, which relate to professional pressures to "turn the page" on the pandemic, the patchy resourcing of new initiatives at the organizational level, and the distraction of health reform.

provided the original author and source are credited.

**Data availability statement:** This paper draws on a qualitative dataset that involved interviews with health system stakeholders involved in the response to the Covid-19 pandemic and its aftermath in Colombia, who represent potentially vulnerable staff at various system levels. The raw interview recordings and transcripts include potentially identifying or sensitive personal and organizational information. Following research ethics guidance relating to participant anonymity, we are not able to disclose the raw data. However, the minimal dataset, with regard to types of interview participant and relevant quotations derived from the interviews, are included in anonymized form in the manuscript. All relevant data are within the manuscript and its Supporting Information files.

**Funding:** ST received funding from "Fondo de Apoyo a Profesores Asistentes" (FAPA) de la Universidad de los Andes. The funders had no role in study design, data collection and analysis, decision to publish, or preparation of the manuscript.

**Competing interests:** I have read the journal's policy and the authors of this manuscript have the following competing interests: ST is an Associate Editor of PLOS ONE. The authors confirm that This does not alter our adherence to PLOS ONE policies on sharing data and materials.

## Introduction

Threatening to overwhelm service capacity and workforce capabilities, the pandemic necessitated decision-making on adaptation at multiple levels of health systems. An international review provided examples of rapid forms of adaptation in response to the pandemic at the micro (professional identities and responsibilities), meso (organizational learning and governance), and macro (regulation, financing and policy) levels [1]. The research literature demonstrates the effects or externalities experienced by health systems from making such rapid adaptations in a context of crisis, with the burden of the response, and negative effects being felt particularly by front-line workers [2–4]. Approaches to learning in this severe context were unusual and remain under characterized. Little is known about how organizations approached learning in this strained context and what influenced the sustained value of such learning. Sustaining innovations has been analyzed from the perspective of stable context conditions, as in the case of new models of stroke care or knowledge translation practices [5,6]. Drawing on recent interview data (2023−24) from stakeholders associated with the Colombian health system, qualitative findings are used to argue that barriers to sustained learning from emergency contexts stack up due to the turbulence of the context associated with the pandemic and its aftermath.

### Conceptualizing organizational learning for system adaptation

The field of organizational learning is saturated with numerous, often competing perspectives on what organizations need to know – and how best to support knowledge formation – to meet public or private interests. An influential theory within this field is associated with cognitive and behavioural perspectives on learning, namely a particular conception of, and pitfalls associated with, experiential learning [7]. This body of work claims that new learning produced through exploration comes from either "organizational search" or "trial-and-error experimentation". Given the need for adaptation to organizational routines demanded by the pandemic, such learning within health systems is likely to have been characterized by an exploratory approach. According to Levitt and March, exploratory learning depends on the operation of three mechanisms: (1) interpreting experiences, (2) encoding those experiences into routines, and (3) evaluating outcomes associated with novel routines. These mechanisms allow organizations to engage in learning by drawing lessons from experiences to inform ongoing behaviour, while "favourable" evaluation of learning experiments is typically needed to facilitate stakeholder agreement on changing organizational course.

Levitt and March's model was chosen to inform this study of post-pandemic learning for various reasons. First, their perspective has been highly influential in the organizational learning field and continues to frame many contemporary approaches to learning. While Levitt and March's classic paper [7], cited over 15,000 times, was published in the late 1980s, a more recent review treats organizational learning in similar conceptual terms: as the conversion of experiences into tacit knowledge that, in turn, becomes embedded in behavioural routines or codified in systems or procedures [8].

Second, Levitt and March's emphasis on collective social routines is relevant to the analysis of learning in health services, as these represent multidisciplinary settings requiring the social coordination of multiple professional groups. For example, analyzing the redesign of an elective surgery pathway in a Norwegian hospital, the significance of modifying routines collectively based on a new paradigm was noted as clinicians acquired an understanding of the wider clinical system in which their roles were situated [9].

Third, Levitt and March underline the importance of the environment or context in shaping learning processes, making their model relevant for analyzing pandemic-related learning that involved responding to a macro environmental shift. Other influential theories of learning place more emphasis on lower-level contextual variables, e.g., individual level learning is the starting point for Nonaka and Takeuchi's knowledge creation spiral (that involves interactions between employees' tacit knowledge) [10] and Argyris' concept of cognitive learning (where an individual's assumptions are said to need updating to address novel problems) [11]. While the environment can stimulate learning processes in these different approaches, Levitt and March's model treats context-specific learning as a collective and distributed practice that is reproduced (or constrained) over time via routines, rather than as a micro level process that begins with cognitive learning by individuals.

In summary, the focus on collective, context-dependent routines makes Levitt and March's model appropriate for this study of learning due to (a) the presence of multiple stakeholders, rather than individual cognitive agents, across health systems that require coordination and (b) the reworking of contextual variables at multiple levels during and following an environmental shock, as represented by the Covid-19 pandemic.

Fig 1 shows a model of the learning process based on Levitt and March's work. The cycle begins with interpreting experiences. In contrast to the normative features of "learning organizations", Levitt and March's perspective highlights the trials of organizational learning: modifying routines is often resisted unless organizations are confronted by "glaring performance errors" [12]. The influence of the context is limited predominantly to signaling underperformance, as such recognition depends on environmental feedback, which shows a need for engaging in exploratory learning processes. Levitt and March recognize that various learning biases moderate the translation of individual and collective experiences within organizations into lessons for guiding future actions, including: interpretation biases (embodied in organizational "stories" or "myths"); institutional interests (governing which experiences are encoded into encoding organizational memory); political claims to success (powerful actors can force reinterpretation of programme objectives or outcomes to align with their interests); and competency traps (organizational inertia can sustain suboptimal routines). These learning biases are likely to affect the use of formal "organizational learning mechanisms" [13] for encouraging improvement.

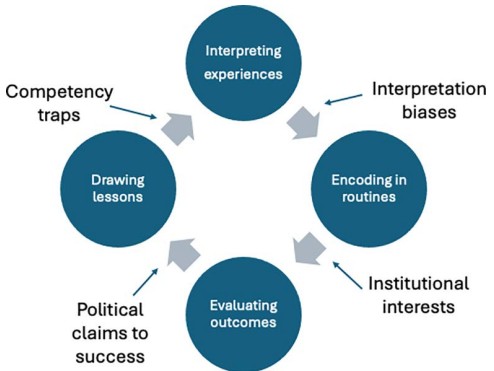

**Fig 1. Elaboration of Levitt and March's model of learning.**

**The turbulent context associated with Covid-19**

Levitt and March's theory describes learning in relatively stable contexts: decision-makers need to evaluate experiences, and associated outcomes, in relation to an established baseline or standards to make choices among alternatives for guiding future behaviour. Their proposed model of learning is readily frustrated by a changing environment, as Levitt and March [7] acknowledge: "Environments change endogenously, and even relatively simple conceptions of learning become complex" (p.331). The assumption of relative environmental stability for the operation of the "interpreting, routinizing and evaluating" experience model is questionable when approaching organizational learning in urgent contexts like pandemics, raising questions about which environmental factors might affect the efficacy of Levitt and March's model in such scenarios. While crisis contexts are more likely to motivate learning [14], such contexts can be characterized by environmental uncertainty and turbulence which degrades the resources available for navigating the process of updating routines [15]. Environments associated with radical uncertainty pose a challenge to the use of existing frames to interpret new experiences, to the updating of routines to match unfamiliar experiences, and to giving sufficient space to outcome evaluation where urgency of change is prioritized.

The turbulent context associated with the pandemic appeared to undermine, or at least reshape, existing institutional structures for guiding learning. First, health systems needed to adopt wide-ranging new habits and routines to respond to the novel threat of Covid-19, including shifts to service priorities, flexible use of financing and regulation, changes to professional roles and relationships, and pursuing new forms of inter-organizational collaboration [1,16]. The context was in a state of flux as changes were needed, and emergency actions hastily undertaken, at multiple levels. Second, health systems' experiences – both historical and emergent – did not represent a ready source of lessons for learning how to tackle the unprecedented threat posed by Covid-19. Health care organizations turned to ad hoc approaches to learning based on trial-and-error [17]. Third, the turbulent context associated with Covid-19 presented challenges for choosing among alternative routines. In particular, difficulties with regard to making time for, and how to approach, the evaluation of outcomes associated with learning by trial-and-error in a context of urgency. Changes in practice initiated hastily may have precluded or constrained evaluation, including through evidence use and stakeholder involvement [18]. In such turbulent contexts, conceptually important sources of learning noted by Levitt and March are potentially diminished.

## Materials and methods

### Research context

**The Colombian health system in the aftermath of the pandemic.** Colombia has a social insurance system, traceable to the Constitutional Act of 1991, which aims to provide universal health coverage [19,20]. The insurance-based system reached 94.7% of the population by 2018 [21]. Colombia fared relatively well compared to other countries geographically nearby, ranking seventh in the Region of the Americas in terms of the number of deaths from COVID-19 by 2021 [20]. Mirroring international experiences [1], early adaptive responses to the pandemic were propelled by organizing around shared purpose, although this motivational effect was moderated in Colombia by excessive centralization and service fragmentation [22]. Lessons included more anticipatory policy learning [23], reducing coordination costs [24], and improving the reimbursement system [25]. Post pandemic challenges include an economic deficit [26] and resourcing [21]: there are 2.5 practicing doctors per 1,000 population (OECD average 3.7); 1.6 practicing nurses (versus 9.2); and 1.7 hospital beds per 1,000 population (versus 4.3).

**National health reform.** In 2023, the national government proposed health reform [27] that questioned the "economic" logic of the current insurance-based system. Proposed changes to promote a "social" logic include: a centralized payment system to ensure the liquidity of health service providers [27]; a transformed role for insurance companies as "managers" who receive a percentage for administration and a bonus for health outcomes [28]; and a network of primary care centres to address social determinants of health [29,30]. The government argues that the proposed reform will initially

incur higher costs, funded through tax reform, due to expanded primary care, formalization of healthcare workers, and improved hospital infrastructure [31]. Longer-term savings from preventative care are anticipated. A stronger state role in coordinating the governance, financing, and delivery of services should improve public oversight [30]; concerns exist regarding risks of corruption in resource management [30] and allocating institutional responsibilities for users' rights [28].

## Conceptualizing learning for this study

This study examines the relationship between turbulent contexts – as characterized by the environment during and after the Covid-19 pandemic – and learning processes within health systems. It is known that interactions between context and experience shape learning processes [8]. This paper examines this relationship by evaluating the relevance of Levitt and March's experiential mechanisms for sustaining learning in the context of the pandemic's aftermath in Colombia. It seeks to understand the relevance of Levitt and March's [7] model of learning as "encoding experiences from history into routines that guide behaviour" (p.320) in turbulent contexts. We hypothesize that the applicability of this model might be reevaluated in relation to this type of context as the mechanisms of learning it posits are blunted by the turbulent context experienced during the pandemic and in its aftermath. Using a series of guiding questions (Table 1), we address the following research question: how do turbulent contexts shape approaches to organizational learning and its sustained value to health systems?

## Data collection

We conducted an interpretivist qualitative study to analyze healthcare stakeholders' perceptions of barriers and enablers for sustained organizational learning post-pandemic. The study was carried out in accordance with the consolidated criteria for reporting qualitative studies [32]. We conducted 21 semi-structured virtual interviews with participants from the Colombian health system (October 2023 – April 2024). We sampled purposively stakeholders able to comment at multiple levels on the Colombian health system in the pandemic's aftermath, including representatives of national government, service providers, administrative staff, clinicians, including physicians and nurses, professional associations, and academics (Table 2). The interviews were conducted by one of the authors (MG) and another researcher (DC, listed in the acknowledgements). Both researchers are PhD students, with general training in social science research methods, who received support from ST on approaching data collection. The interviewees were contacted for the first time (typically via email) by the researchers about participating in the study. The interviews lasted 45 minutes on average, were conducted virtually (e.g., Microsoft Teams), and were semi-structured using a topic guide. The interviews are complemented with one participant's contribution to one of six focus groups during an earlier phase of our study (2021). All study participants received an information sheet from the interviewer that explained the goals of the study; participants then provided informed consent. All interviews were transcribed professionally. Ethical approval (low risk) was obtained from the University of los Andes.

**Table 1. Guiding questions and link to study design.**

| Guiding question | Link to study design |
|---|---|
| How are approaches to organizational learning shaped by turbulent contexts? | Investigate changes to the context at multiple levels (micro, meso, macro) and analyse how these affect approaches to learning. |
| What is the relevance of Levitt and March´s model in explaining learning associated with the pandemic? | Review interviewees´ narratives of learning and innovation with regard to the relevance of interpreting experiences, codifying routines or new ways of working, and evaluating outcomes. |
| What is the sustained value of new routines developed in response to turbulent contexts? | Trace prospects of learning and innovations initiated during the pandemic and analyse barriers and enablers to their contemporary use |

**Table 2. Characteristics of the interview participants.**

| Code | Interviewee description | Level | Date |
|------|------------------------|-------|------|
| NP1 | Professional Association | Meso | Jan 2024 |
| NP2 | Public provider | Meso | Jan 2024 |
| NP3 | Private provider | Macro | Oct 2023 |
| NP4 | NGO | Micro/ macro | Jan 2024 |
| NP5 | Government department | Macro | Nov 2023 |
| NP6 | NGO | Meso | Oct 2023 |
| NP7 | NGO | Micro/ meso | Oct 2023 |
| NP8 | Insurance company | Micro/Meso | Oct 2023 |
| NP9 | Professional Association | Micro | Feb 2024 |
| NP10 | Professional Association | Micro/Meso | Nov 2023 |
| NP11 | University | Macro | Nov 2023 |
| NP12 | Government department | Macro | Feb 2024 |
| NP13 | Insurance company | Meso | Feb 2024 |
| NP14 | NGO | Meso | March 2024 |
| NP15 | Pharmaceutical | Macro | March 2024 |
| NP16 | Government department | Meso | March 2024 |
| NP17 | Private provider | Micro | March 2024 |
| NP18 | Private provider | Macro | March 2024 |
| NP19 | Innovation centre | Meso | April 2024 |
| NP20 | Professional association | Meso/Macro | April 2024 |
| NP21 | Public-private provider | Meso | April 2024 |
| BogAc1102 | Professional association | Meso | Nov 2021 (focus group) |

## Data analysis

The research follows an abductive approach for empirically based theory construction, which considers the existence of a theoretical background but allows the emergence of new theoretical insights [33]. In the data analysis we extracted relevant inductive themes from the interviews while considering a broad deductive theoretical framework about learning in organizations. The concepts framing data analysis are derived from the mainstream theory of learning in the cognitive and behavioural tradition of Levitt & March [7]. In relation to our analytical focus on barriers and enablers for learning after the pandemic, abductive analysis involved the following stages:

1. Each researcher (ST and MG) initially coded three to five interviews independently and proposed potential codes for analyzing the subsequent interviews.

2. Similar codes were identified and grouped in subthemes.

3. Subthemes of analysis were categorized according to their relation to macro, meso, or micro contextual levels of analysis.

4. Three construct concepts were developed to summarize the main findings at the micro, meso and macro levels.

5. Any differences in interpretation were resolved through debate.

Findings were contrasted with organizational theory concepts to extend and further develop Levitt & March's three main learning mechanisms: (1) interpreting experiences, (2), encoding those experiences into new or updated routines, and (3) evaluating outcomes associated with novel routines [7]. Quotations from the participants are used to illustrate the themes. The identities of participants were anonymized and replaced with a participant code.

## Results

### Micro level

Socio-psychological barriers and staff churn contributed to what we term a "forgotten context" at the micro level, especially in front-line operational areas, where our interviews suggested that opportunities for processing pandemic experiences to guide learning in the pandemic's aftermath were limited. The capacity of health professionals to learn from the pandemic included functional and psychological components. We describe the pandemic as a forgotten context due to diminished human resource capacity and "active forgetting" of traumatic experiences, along with inconsistent use of mental health interventions. Functionally, the pandemic impacted workforce capacity and training. Staff numbers increased during the pandemic but dwindled afterward due to redundancies, international migration for better conditions, and personal career changes, affecting organizational memory from this period:

> "What became clear during the pandemic was that many employees were laid off because, since it was a service provision, without requests for medical procedures, "you're surplus, out you go," and a lot of people were let go, there was a loss of jobs in the healthcare sector at that time..." (NP01, Professional Association).

The pandemic's impact on professional training quality affected new medical professionals, referred to as "post-pandemic doctors", who relied more on virtual training, reducing their "real-world" learning exposure in care settings:

> "…what we're receiving now are medical professionals who are quite detached from the real needs of the services, at least in our emergency department. This virtual setup distanced them from the real-world training that a healthcare professional should have, you know…we call them "post-pandemic" doctors because they're the ones who came after" (NP17, Private provider).

The pandemic was a source of workforce trauma that had persisted for many staff. The following interviewee describes the severe mental health impact of the pandemic on many health care personnel, notably among those delivering front-line care to severely ill patients and feeling implicated in the deaths of patients. Traumatic experiences persisted through a continuing "feeling of guilt":

> "The mental health impact was very severe on healthcare personnel... I had colleagues, nurses, doctors, who would tell me: It's difficult to decide about a patient's life when you have 10 patients outside struggling to breathe, and everything inside is already full. So, if one dies, one dies, but it's one more ventilator and one more bed I have to save a life out there. Who do I prioritize? Who do I give a chance to live? I think that marked them a lot. There are people who still carry that feeling of guilt…they still question themselves, maybe because so many people died and they could have done more" (NP07, NGO).

Our study revealed that many health system stakeholders either avoided discussing pandemic trauma or lacked opportunities to do so. Some showed disinterest in recalling or learning from the pandemic. One interviewee noted that enthusiasm for mental health initiatives waned due to pressure to "turn the page" on the pandemic:

> "there were many campaigns during the pandemic to take care of the mental health of health personnel, there were many, but now it is a topic that was not talked about again, in fact the pandemic in general is a topic that no one wants to talk about anymore. Everyone is already avoiding it, in fact, if you see a book or a movie about a pandemic, you get past it and that's the feeling I have, that this is how many people feel who already want to turn the page on the subject of the pandemic and everything that was associated with it" (NP04, NGO).

Opportunities for processing trauma from the pandemic had reportedly diminished. Multiple factors worked against dwelling on the trauma of pandemic-related experiences, including stoic professional identities, the unrelenting tempo of medical work, and institutional encouragement to return to productive working practices. As the following interviewee remarked, these pressures to "move on" had a negative impact both on personal mental health and on the quality of inter-actions among staff: "*you had to face the pandemic, you had to face things you didn't know, you acted, you worked, you kept your promise to act for the lives of others, and now that's it, move on and goodbye*" (NP 21, Public-private provider).

**Meso level**

At the organizational level, a "stretched context" was identified whereby experiences during the pandemic had a con-tinuing influence on organizational practices during its aftermath, especially among service providers. Key mechanisms included leaders' use of motivational story-telling that referenced the pandemic, while implementation constraints were encountered in the current context that were traceable to the pandemic. The findings at the meso level point to barriers to sustaining technical innovations due to several aspects: (1) infrastructural barriers like ongoing ownership and budget-holding, (2) the cost of new epidemiological demands at the population level, and (3) a dearth of evaluative evidence. Social practices were perceived to have been easier to sustain via leaders' stories of accomplishment that recount, and aim to reaffirm, workforce resilience and inter-unit cooperation from during the pandemic era.

Disagreements over organizational roles and funding hindered the continuation of technical innovations post-pandemic. For example, one interviewee explained that learning from the use of telemedicine had not been utilized fully because of a lack of financial commitment on the part of insurers and government, suggesting regression or "backwardness" in relation to these services in the pandemic's aftermath:

"*A budget was needed, so the budget had to go to the mayor's office or it had to come from the national budget and there was not, and then the differences began and on the other hand the [insurance company] did not want to promote telemedicine either, because that was a job that also corresponded to them, so that management of primary care, which corresponds partly to the government and for the most part to the [insurer], that was not done and we were not prepared for telemedicine to be able to reach many more people. That has been a backwardness, I would believe what this could have taught us, however, today we do not take advantage of it*" (NP02, Public provider).

Second, new health service costs, and demands on clinicians, had emerged from managing the population health effects of the pandemic. A new epidemiological profile resulting from the pandemic, whereby patients were presenting with higher disease severity and co-morbidities, had unexpectedly consumed health care organizations' resources:

"*From a sustainability and financial perspective, the pandemic itself generated a multitude of additional effects beyond Covid-19. Covid brought along with it other diseases, essentially altering the epidemiological profile, which has been quite significant and has also changed the way we view and understand costs and the healthcare model, among other things... People stopped consuming healthcare services for a while, and when they returned, they returned in some cases with a higher level of severity, with a much more complicated state of the disease*" (NP13, Insurance Company).

Emergency regulations for workplace adaptation expired, including temporary financing and resources, affecting ongoing safety measures. Biosecurity, critical during the pandemic, saw diminished enforcement along with the focus on address-ing future pandemic risks, as noted by an interviewee:

"*Nothing, I mean nothing, not even occupational health and safety programs foreseeing a new pandemic. One would think, Let´s prepare ourselves, let's establish norms regarding human resources so that what we experienced doesn't*

*happen again. But there's zero, absolutely zero, in terms of occupational health and safety programs. It's as if nothing happened"* (NP16, Government department).

Fourth, decision-making on sustaining service innovations sometimes suffered from a lack of evaluative evidence as changes were initiated hastily during the pandemic. The consistently high pace of service changes demanded during the pandemic had precluded comprehensive documentation, leaving a legacy of unsubstantiated "anecdotes" rather than reliable evidence as the following stakeholder explained:

*"We were talking about intensive care units, ventilators, deaths, then we are talking about vaccines, we are no longer talking about vaccines but about vaccinated, not about vaccinated but about population with the third doses. The change caused by the pandemic is so rapid that we have to deal with what happens today and yesterday is going unsupported, and what remains undocumented becomes an anecdote"* (BogAc1102, professional association, 2021 focus group).

There were accounts of sustained learning from the pandemic, especially in relation to social practice innovations like improved teamwork. Stories of accomplishment from the era of the pandemic were reportedly retold by health system leaders to shape orientations among the workforce toward ongoing learning and change. For example, this interviewee with a leadership role within an insurer explained how tales of accomplishment around "handling" challenges during the pandemic had been used to influence the attitudes of staff to new operational and strategic challenges, suggesting that such achievements improved the cohesiveness of professional teams: "*Another issue is team cohesion; in almost all teams, you hear things like: well, if we could handle this during the pandemic, then we can handle this now*" (NP13, Insurance Company).

Another interviewee explained how staff had become "more open to change" following experiences during the pandemic, including increased willingness to rotate among different departments which had been a necessity at that time: "*they realized the importance of being prepared to rotate to any service they're needed in... they're not as closed off to the idea*" (NP07, NGO).

### Macro level

At the macro level, our interviews suggested an ongoing "volatile context" exists as proposed health reform turns attention away from post-pandemic learning. For some interviewees, reform negates potential for learning by overlooking some pandemic lessons and focusing actors on new priorities and uncertainties. In February 2023, a radical health reform proposal was published [27]. Our analysis of the 278-page reform bill found it draws lessons from the pandemic about improving public health and prevention, addressing deficiencies in information systems, and recognizing regional inequalities in health service access and outcomes. Implementing these national-level realizations should improve preparedness for future emergencies, as suggested by the following interviewee:

*"the health system has improved in that we realized, or the entire administrative part realized, that we have to be prepared for events like this, that we have to pay a little more attention to health prevention, to primary care, to the availability of supplies, of equipment, of technology"* (NP18, Private provider).

However, our recent round of interviews suggested that other potential lessons from the pandemic had been overlooked in the reform bill, as "ideology" had overshadowed "evidence". Insurers that were key to the pandemic response faced restructuring:

*"We have not had this learning because we do not learn from evidence. There is a political management of health, a collusion to manage the politics of fear, of restrictions. The logic of fear empowers governments and fosters uncertainty.*

*These changes in the health system are political changes, not technical-scientific ones. Therefore, we are not thinking about science. There is a disdain for science, a disdain for evidence because activism has prevailed, because ideology has been placed above science. We are living in an era of ideology"* (NP03, Private provider).

One of the interviewees highlighted the role played by insurance companies in areas such as contracting, risk management, equipment procurement, and service provision, expressing concern about the state's capacity to assume these tasks. As a result, there is a risk that the lessons learned from the pandemic regarding these issues will not be sustained due to the role changes proposed by the reform:

*"a very important north is lost with eliminating many of the actors that were [important] during the pandemic, the [insurance companies] were very important; one may have a negative image of them or whatever, but they played a very important role in risk management and in the issue of vaccination, they were the main ones. They were there, they were in the care of patients, and that same autonomy they had made the system very flexible in contracting, in the purchase of medical equipment and, now with a centralized system, because I see it as too complex, I don't see the capacity in the state to address such everyday issues as purchasing* (NP04, NGO).

Another interviewee evaluated the health system's performance during the pandemic as acceptable but believed the reform is "sabotaging" the role of insurance companies. This quote suggests concern that the lessons learned from a health system "put to the test" during the pandemic may not be sustained:

*"…the healthcare system was put to the test during the pandemic, but it managed to function despite everything. Everyone had the opportunity to receive treatment. At this moment, I think we are all observing with great concern so many reforms, especially those that seem like significant sabotage to [insurance companies]"* (NP11, university).

Moreover, the radical nature of the reforms proposed, notably a stronger state role in the regulation and delivery of health services, was seen by some of our interviewees as a distraction that was affecting demand in the incumbent system (e.g., patients' hoarding medications) and workforce preferences due to the uncertainty related to job continuity: *"...there are many doctors who are scared, there is a lot of uncertainty, that is absolutely clear with the issue of the reform"* (NP15, Pharmaceutical).

## Discussion

This paper highlights trials of sustaining learning from experiences of health systems gained during emergency contexts like pandemics. Table 3 summarizes the results by analytic level. Barriers to sustained learning originate at multiple health system levels. At the professional level, workforce capacity to sustain learning is limited by staff churn during and after traumatic events, affecting organizational memory. Unaddressed occupational disease, that manifests as stress and burnout [34], stifles motivation when staff confront new task demands. At the organizational level, social practice innovations, including improved collaboration and learnt resilience in the face of new challenges, were reportedly easier to sustain than technical service changes that required decision-making about post-pandemic ownership, financing, and evaluation of their ongoing relevance. At the macro level, the political cycle was found to pose an obstacle to lesson drawing from the pandemic, with health reforms in Colombia perceived by some to privilege ideological concerns over evidence. Moreover, priorities for reform had distracted some actors' attention away from using lessons to improve the incumbent system. This study extends Levitt and March's organizational learning framework in relation to turbulent contexts by introducing additional factors that moderate the mechanisms of learning they describe.

**Table 3. Summary of results by analytic level.**

| Level | Theme category | Sub-themes |
|---|---|---|
| Micro | • Forgotten context | • Diminished workforce capacity<br>• Quality of "virtual" professional training<br>• Barriers to addressing mental health challenges in the workforce include stoic professional identities, the unrelenting tempo of medical work, and institutional encouragement |
| Meso | • Stretched context | • Stories of accomplishment as leadership device (e.g., social practice innovations and inter-unit cooperation)<br>• Financial barriers to sustaining technical innovations (e.g., telemedicine)<br>• Period of emergency regulation to facilitate adaptation at meso level passed<br>• Lack of evaluation undermines translation of experience into knowledge (i.e., can remain anecdotal evidence) |
| Macro | • Volatile context | • Reform can overlook potential lessons from the pandemic<br>• Reform occupies actors' attention with new priorities and forms of uncertainty |

## Research implications

In cognitive and behavioural theories of the firm, the context or environment is regarded as a trigger of, and resource for, organizational learning (e.g., environmental change can prompt the search for new routines, while environmental feedback from trial-and-error experiments can guide future actions). It is known that experiential learning processes, from search to knowledge creation, retention, and transfer, can be inhibited or facilitated by the context [5]. However, less is known about how the context affects the sustained value of lessons drawn from experiential learning. In processes of sustained learning – or forgetting – this study suggests that the receptivity of the context to learning is not fixed; rather, it has malleable properties that affect learning which shift over time. We have updated our depiction of Levitt and March's model of learning to reflect our findings on sustaining learning in conditions of environmental turbulence (Fig 2). The cycle begins with context turbulence, which is a trigger for reinterpreting experiences.

We suggest that context turbulence is recognized as an additional mediating variable in learning processes. A turbulent context can simultaneously trigger learning processes (i.e., problem-driven search), while being a precursor to how experiences are interpreted, as the context embodies existing institutional interests and earlier rounds of routines. The influence of macro level environmental change on learning processes at the micro and meso levels found within our study has led us to question the value of terming environmental change as "endogenous" in Levitt and March's model. While using the term endogenous rightly points to sources of change associated with the environment that require organizational responses, those changes become entangled with learning processes at the micro and meso levels over time. Shifting system priorities associated with reform not only affect the day-to-day learning processes of actors at lower levels (e.g., making sense of new priorities that take over, and potentially distract from, consolidating existing lessons), they also influence actors' strategic interests as they aim to reshape reform processes (e.g., insurers' political lobbying for system maintenance by engendering recognition of the role they did play in responding to the pandemic). Over time, the supposed endogeneity of environmental change becomes questionable as the environment informs, and evolves in dynamic interaction with, learning processes at other levels of the health system.

Turbulent contexts trigger and mediate the relationship between experiences and lessons. Environmental turbulence should be treated as a fluid construct that influences, and interacts with, the sustaining of learning processes over time. In Colombia, that fluidity was expressed in the context sometimes being "forgotten", due to workforce churn and trauma, affecting organizational memory; sometimes being "stretched" as war stories from the pandemic of adaptive resilience and teamwork, on the one hand, or deficits in implementation resources on the other, persist post pandemic; and sometimes

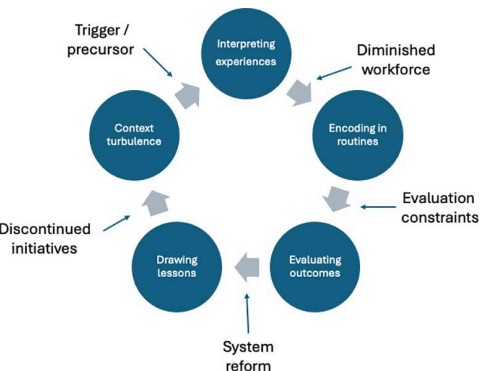

**Fig 2. Addition of context turbulence to Levitt and March's model of learning.**

"volatile" as shifting priorities associated with the political cycle become a distraction from, and potentially take priority over, the application of lessons past in incumbent systems. We suggest that the macro environment should be recognized as a mediator of organizational learning that shapes the very experience of organization, and the subsequent steps of interpreting, routinising and evaluating those experiences which involve ongoing interactions with the environment. Below we expand on the influence of these additional moderating variables at different stages of Levitt and March's organizational learning process.

**Interpreting experiences.** Organizations' ability to interpret experiences to guide future actions is mediated by human resource capacities, including workforce mental health, for ongoing learning and implementation during and post crisis. Making sense of experiences relies on capable and adequately supported human resources. While Levitt and March recognize the socio-psychological basis of learning, including interpretative biases, a normative motivation to learn is assumed among human agents (albeit they acknowledge "competency traps" that dissuades further exploration). Our study suggests that the capacity of human resources, including staff burnout and churn that hinder further rounds of learning in the aftermath of crises, needs to be taken into account when conceptualizing sustained learning. Behavioural theories of organizational learning could be usefully combined with human resource management research that takes a work and organizational psychology perspective [35], allowing a finer-grained understanding of the workforce to explain differences between the "intended" and "implemented" aspects of human resource interventions directed towards learning, especially following exposure to trauma in health care settings.

**Encoding into routines.** Encoding those experiences into routines relies, yes, on telling stories of accomplishments as Levitt and March argue, but the motivational aspect of sharing "war stories" should be matched with resources for implementation to sustain innovative routines. Corporate "stories" or "myths" that are said to encapsulate learning mean little in the absence of a supportive context for translating leaders' motivating words into actions that can be feasibly taken by their staff (i.e., giving staff adequate training and resources for implementing desired changes in ways of working). Such implementation considerations should temper claims about "political" leadership [36] embodied in personal traits including negotiation and persuasion for fostering organization-wide learning – especially in the aftermath of crises – as enacting change rests on the workforce who require support with implementation that goes beyond motivational story-telling.

**Evaluating outcomes.** The ability to evaluate outcomes associated with new or modified routines in turbulent contexts rests on building in time for reliably assessing alternatives (e.g., use of rapid evaluation frameworks). In turbulent contexts, evaluative processes are constrained by urgent pressures to make rapid service adaptations, which not only limits the time available to set up and maintain evaluation of service change, but the characteristics of sustained environmental

turbulence – from ongoing ambiguity to volatility [25] – undermine both the generation of appropriate actions for tackling shifting problems and the measurement of associated outcomes. Environmental flux moderates robust evaluation of adaptive change.

**Drawing lessons.** Translating experiences into lessons presupposes continuity of the system from which lessons are derived. Drawing lessons relies on system maintenance. The macro political context tends to have a short cycle (e.g., presidents are limited to four years in Colombia), meaning that lessons from history tend to be short-lived as shifting political ideologies reframe or even overlook past experiences. Gustavo Petro, regarded a left-wing president, took over from successive right-wing presidents of Colombia in August 2022. Health reforms, led by Petro, embody a particular political ideology (i.e., a return to the state) that implies significant restructuring of the "managed competition" model [37] that has prevailed in Colombia under right-leaning governments since the early 1990s. Potential lessons from the pandemic have undergone political filtering. The macro political context shapes learning and promotes discontinuity of initiatives at lower organizational levels by creating uncertainty and distraction among managers and professionals.

## Policy and practice implications

Implications for enabling organizational learning centre on acknowledging the mediating role of environmental turbulence and on addressing contextual moderators of learning processes. With regard to tackling moderating factors, health service managers should recognize the critical role of human resources, and investing in a durable infrastructure, to support and sustain service change. Relevant interventions include provision of psychological support programmes for targeting unaddressed mental health issues, including "burnout" as a recognized occupational disease [34], and investing in a supportive work environment to help prevent the realization of acute workforce risks (e.g., maintaining organizational slack and involving the workforce in decisions about service change). Strategic planning should span political cycles to give sufficient space for lesson drawing from past experiences.

## Conclusion

Policymakers should place emphasis on anticipatory planning for future crises. While Levitt and March [7] argue that "routines are based on interpretations of the past more than anticipations of the future" (p.320), it is important that clear pathways are identified and maintained for applying lessons drawn from the past to improve responses to anticipatable aspects of future pandemics. A key strength of this study is the use of longitudinal data that drew on qualitative data collected during and in the aftermath of the pandemic. Successive rounds of data collection allowed us to capture different ways in which the evolving context interacted with learning processes during and in the aftermath of Covid-19. The study is limited by use of a single case. While this provided a deep understanding of learning processes within and across one health system, further research could usefully differentiate among approaches to learning – and their sustained value for health systems – by service area, provider type, and national setting. Furthermore, the study lacked access to some stakeholders, such as local government agencies and patients' associations, who could have contributed valuable perspectives to the study.

## Acknowledgments

We wish to thank David Carrasquilla for his support with project management and data collection. We also acknowledge the interview/ focus group participants for giving up their valuable time and insights to support this study.

## Author contributions

**Conceptualization:** Simon Turner.

**Data curation:** Mary Ruth Guevara Maldonado.

**Formal analysis:** Simon Turner, Mary Ruth Guevara Maldonado.

**Funding acquisition:** Simon Turner.

**Investigation:** Simon Turner, Mary Ruth Guevara Maldonado.

**Methodology:** Simon Turner, Mary Ruth Guevara Maldonado.

**Project administration:** Simon Turner.

**Supervision:** Simon Turner.

**Writing – original draft:** Simon Turner, Mary Ruth Guevara Maldonado.

**Writing – review & editing:** Simon Turner, Mary Ruth Guevara Maldonado.

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
