## [Decision Letter · Decision Letter 0]

PONE-D-24-34928How can health systems sustain lessons drawn from emergency contexts? Evidence from ColombiaPLOS ONE?

Dear Dr. Turner,

Thank you for submitting your manuscript to PLOS ONE. After careful consideration, we feel that it has merit but does not fully meet PLOS ONE’s publication criteria as it currently stands. Therefore, we invite you to submit a revised version of the manuscript that addresses the points raised during the review process.

Please follow the Reviewer’s recommendations, especially in terms of the length and organization of the manuscript, and justify your decision to employ Levitt and March’s perspective.

We look forward to receiving your revised manuscript.

Kind regards,

Ietza Bojorquez, Ph.D.

Academic Editor

PLOS ONE

“ST received funding from “Fondo de Apoyo a Profesores Asistentes” (FAPA) de la Universidad de los Andes”

“I have read the journal's policy and the authors of this manuscript have the following competing interests: ST is an Associate Editor of PLOS ONE.”

4. We noted in your submission details that a portion of your manuscript may have been presented or published elsewhere. [Yes, as explained in the cover letter, a related manuscript is under review in this journal. However, the current paper draws on a distinctive dataset that was conducted later (2023-24), original conceptualisation, and analytic perspective. There is no overlap between the two papers. For transparency, the paper under review with PLOS ONE is included with the submission as a supplementary file.] Please clarify whether this [conference proceeding or publication] was peer-reviewed and formally published. If this work was previously peer-reviewed and published, in the cover letter please provide the reason that this work does not constitute dual publication and should be included in the current manuscript.

5. We note that you have indicated that there are restrictions to data sharing for this study. PLOS only allows data to be available upon request if there are legal or ethical restrictions on sharing data publicly. For more information on unacceptable data access restrictions, please see http://journals.plos.org/plosone/s/data-availability#loc-unacceptable-data-access-restrictions.

6. Please include a separate caption for each figure in your manuscript.

7. We note that there is identifying data in Table 2. Due to the inclusion of these potentially identifying data, we have removed this file from your file inventory. Prior to sharing human research participant data, authors should consult with an ethics committee to ensure data are shared in accordance with participant consent and all applicable local laws.

-Location data

Please remove or anonymize all personal information, ensure that the data shared are in accordance with participant consent, and re-upload a fully anonymized data set. Please note that spreadsheet columns with personal information must be removed and not hidden as all hidden columns will appear in the published file.

Reviewers' comments:

Reviewer's Responses to Questions

**Comments to the Author**

1. Is the manuscript technically sound, and do the data support the conclusions?

Reviewer #1: Yes

2. Has the statistical analysis been performed appropriately and rigorously?

Reviewer #1: N/A

3. Have the authors made all data underlying the findings in their manuscript fully available?

Reviewer #1: No

4. Is the manuscript presented in an intelligible fashion and written in standard English?

Reviewer #1: Yes

Reviewer #1: Dear authors,

I appreciate the opportunity to read and contribute to the improvement of this article. This manuscript provides an interesting insight into the learning process and Levitt and March's perspective on this area. The analysis that the Covid-19 pandemic may have reconfigured this 4-step learning cycle is interesting.

At the same time, I would like to make some suggestions that would improve the manuscript:

1. from my point of view, the manuscript is much too long in terms of pages, which makes it difficult to follow the ideas and read it in its entirety. i recommend shortening it where possible, especially those paragraphs that may not be of significant importance to the research being conducted.

2. i think it should be presented that the chapters between "introduction" and "materials and methods" are subchapters related to the introduction chapter. otherwise it is not understandable where they are located. (this also applies to the other chapters)

3. it is not clear why is used the Levitt and March model as a key element, what makes it special to the detriment of other models.

4. in the "data collection" sub-chapter, I would present table 2 from the micro/meso/macro perspective of the respondents. this will allow for a clearer analysis in the results chapter.

5. in the results chapter I would put table 3 at the end of the chapter (as a conclusion). I do not see the need to present the interviewees' statements. It would be more interesting to present similar indicators and differences found at the 3 levels. The individual statements represent a subjective perspective of the subject rather than a general perspective of the level to which it belongs (micro/ meso/ macro).

6. I would recommend integrating Figures 1 and 2 in the text and not at the end (at the same time showing in the diagrams which is the first element from which the cycle starts).

7. some ideas and elements are repeated in the "Results" and "Discussion" chapters, I recommend keeping them in one chapter or combining the two chapters into one.

**Do you want your identity to be public for this peer review?** For information about this choice, including consent withdrawal, please see our Privacy Policy

Reviewer #1: No

---

## [Editor Report · Decision Letter 1]

How can health systems sustain lessons drawn from emergency contexts? Evidence from Colombia

PONE-D-24-34928R1

Dear Dr. Turner,

We’re pleased to inform you that your manuscript has been judged scientifically suitable for publication and will be formally accepted for publication once it meets all outstanding technical requirements.

Kind regards,

Ietza Bojorquez, Ph.D.

Academic Editor

PLOS ONE

---

## [Editor Report · Acceptance letter]

PONE-D-24-34928R1

PLOS ONE

Dear Dr. Turner,

I'm pleased to inform you that your manuscript has been deemed suitable for publication in PLOS ONE. Congratulations! Your manuscript is now being handed over to our production team.

Kind regards,

on behalf of

Dr Ietza Bojorquez

Academic Editor

PLOS ONE